# Analysis of inositol phosphate metabolism by capillary electrophoresis electrospray ionization mass spectrometry

Danye Qiu [1✉], Miranda S. Wilson[2], Verena B. Eisenbeis [1], Robert K. Harmel[3], Esther Riemer [4],
Thomas M. Haas[1], Christopher Wittwer[1], Nikolaus Jork[1], Chunfang Gu[5], Stephen B. Shears [5], Gabriel Schaaf[4],
Bernd Kammerer[1], Dorothea Fiedler[3], Adolfo Saiardi [2✉] & Henning J. Jessen [1,6✉]

The analysis of *myo*-inositol phosphates (InsPs) and *myo*-inositol pyrophosphates (PP-InsPs) is a daunting challenge due to the large number of possible isomers, the absence of a chromophore, the high charge density, the low abundance, and the instability of the esters and anhydrides. Given their importance in biology, an analytical approach to follow and understand this complex signaling hub is desirable. Here, capillary electrophoresis (CE) coupled to electrospray ionization mass spectrometry (ESI-MS) is implemented to analyze complex mixtures of InsPs and PP-InsPs with high sensitivity. Stable isotope labeled (SIL) internal standards allow for matrix-independent quantitative assignment. The method is validated in wild-type and knockout mammalian cell lines and in model organisms. SIL-CE-ESI-MS enables the accurate monitoring of InsPs and PP-InsPs arising from compartmentalized cellular synthesis pathways, by feeding cells with either [$^{13}C_6$]-*myo*-inositol or [$^{13}C_6$]-D-glucose. In doing so, we provide evidence for the existence of unknown inositol synthesis pathways in mammals, highlighting the potential of this method to dissect inositol phosphate metabolism and signalling.

[1] Institute of Organic Chemistry, University of Freiburg, Albertstr. 21, 79104 Freiburg, Germany. [2] Medical Research Council, Laboratory for Molecular Cell Biology, University College London, London WC1E 6BT, UK. [3] Leibniz-Forschungsinstitut für Molekulare Pharmakologie, Robert-Rössle-Str. 10, 13125 Berlin, Germany. [4] Institute of Crop Science and Resource Conservation, Department of Plant Nutrition, Rheinische Friedrich-Wilhelms-University Bonn, 53115 Bonn, Germany. [5] Signal Transduction Laboratory, National Institute of Environmental Health Sciences, National Institutes of Health, Research Triangle Park, NC 27709, USA. [6] CIBSS - Centre for Integrative Biological Signalling Studies, University of Freiburg, 79104 Freiburg, Germany. ✉email: danyeqiu@gmail.com; a.saiardi@ucl.ac.uk; henning.jessen@oc.uni-freiburg.de

**M**yo-Inositol (inositol hereafter) phosphates (InsPs) are second messengers involved in signaling processes in eukaryotes[1]. In principle, 63 phosphorylated InsPs can be generated by sequential phosphorylation of the OH groups of inositol, resulting in significant analytical ramifications. Moreover, the fully phosphorylated inositol hexakisphosphate InsP6, the usually most abundant species, can be further phosphorylated to diphospho-inositol polyphosphates (PP-InsPs), called inositol pyrophosphates (Fig. 1a and Supplementary Fig. 1 for the

mammalian PP-InsPs pathway)[2,3]. These structures contain one (PP-InsP5) or two ((PP)2-InsP4) phosphoric anhydride (P-anhydride) bonds in addition to the phosphate esters (Fig. 1a). The current model for biologically relevant isomers places the P-anhydrides in defined positions. For example, mammals, yeast, and plants produce 5-PP-InsP5 as the main isomer, with the P-anhydride residing in the plane of symmetry at the 5-position[2,4]. The second, lower abundance, isomer is 1-PP-InsP5, which has a biologically likely irrelevant enantiomer, its mirror-image 3-PP-

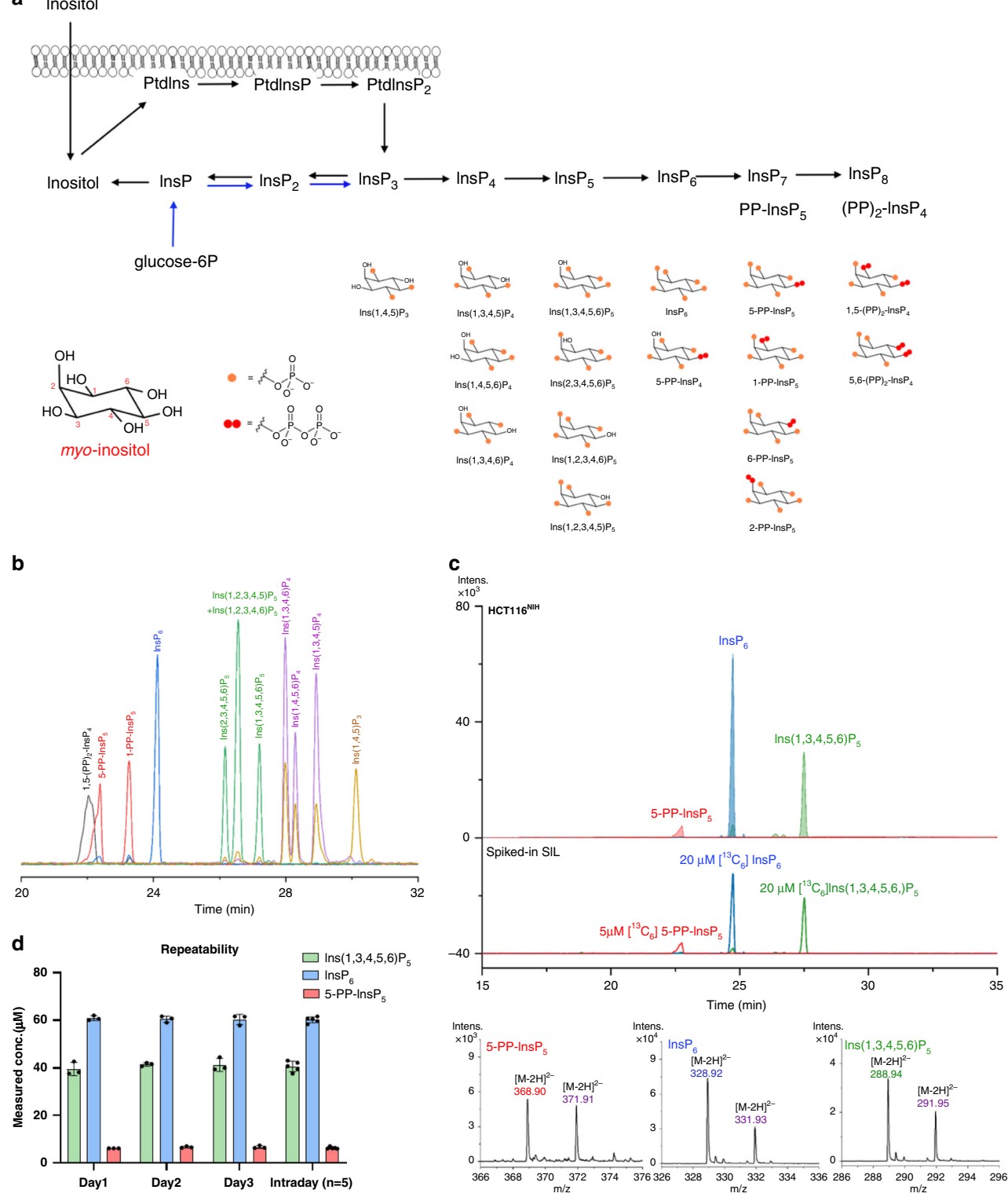

**Fig. 1 Separation of PP-InsP and InsP by CE-ESI-MS. a** Simplified biosynthesis of inositol phosphate (InsPs) and inositol pyrophosphates (PP-InsPs) from lipid PInsP$_2$ and glucose-6P, with the structures of InsPs and PP-InsPs that can be currently resolved by CE-ESI-MS. The metabolic pathways for inositol (pyro)phosphates in mammals are shown in Supplementary Fig. 1. **b** Separation of PP-InsP and InsP standards by CE-ESI-MS. BGE: 35 mM ammonium acetate titrated with ammonium hydroxide to pH 9.7, CE voltage: 30 kV, CE current: 23 µA, injection: 50 mbar, 10 s (10 nL), solutes: 5 µM for Ins(2,3,4,5,6)P$_5$, Ins(1,3,4,5,6)P$_5$, Ins(1,2,3,4,6)P$_5$, Ins(1,2,3,4,5)P$_5$, InsP$_6$, 5-PP-InsP$_5$, 1-PP-InsP$_5$, and 1,5-(PP)$_2$-InsP$_4$; 20 µg/mL for Ins(1,3,4,5)P$_4$ and Ins(1,3,4,6)P$_4$, 8 µg/mL for Ins(1,4,5,6)P$_4$, and 10 µg/mL for Ins(1,4,5)P$_3$. **c** Extracted ion electropherograms (EIEs) of the main inositol (pyro)phosphates in HCT116$^{NIH}$ and spiked SIL IS 20 µM [$^{13}C_6$]Ins(1,3,4,5,6)P$_5$, 20 µM [$^{13}C_6$]InsP$_6$ and 5 µM [$^{13}C_6$] 5-PP-InsP$_5$. The InsP species Ins(1,3,4,5,6)P$_5$, InsP$_6$, and 5-PP-InsP$_5$ in HCT116$^{NIH}$ were assigned by their accurate mass, and identical migration time with spiked SIL standards. **d** Intra- and interday repeatability of the inositol (pyro)phosphate analysis by CE-ESI-MS; data are presented as means ± SD (intraday $n = 3$, interday $n = 5$, independent technical experiments). Source data are provided as a Source Data file.

InsP$_5$. Further phosphorylation of 5-PP-InsP$_5$ leads to 1,5-(PP)$_2$-InsP$_4$. PP-InsPs are considered metabolic messengers, whose functions have recently become the focus of intense research[5]. Across species, they are signals in diverse processes including the regulation of energy metabolism and phosphate homeostasis[6–8]. Other organisms, e.g. the social amoeba *Dictyostelium discoideum*, produce distinct PP-InsP isomers (Fig. 1a) whose functions remain elusive[9,10].

The metabolic complexity of InsP turnover and their low abundance, in combination with the absence of a chromophore, and high charge density, have hampered research into these signaling molecules. The most widely applied quantitation technology relies on metabolic labeling of cells using tritiated [$^3$H]-inositol, followed by acidic extraction, strong anion exchange high performance liquid chromatography (SAX-HPLC), and manual scintillation counting of individual fractions[11]. While this approach is sensitive, it requires a dedicated radioactive suite, is expensive, and labor and time consuming. Moreover, it is blind to inositol generated endogenously from D-glucose-6-phosphate by inositol-3-phosphate synthase 1 (ISYNA1). Therefore, post-column derivatization UV-detection approaches have been developed to avoid radiolabeling[12,13].

Recently, it was shown that inositol tetrakisphosphate 1-kinase 1 (ITPK1), which is present in Asgard archaea, social amoeba, plants, and animals, sequentially phosphorylates Ins(3)P ultimately leading to InsP$_6$ and PP-InsPs[4,12,14]. Different InsP pools generated from exogenously acquired or endogenously synthesized inositol can potentially be monitored using chromatography coupled to mass spectrometry (LC-MS) after feeding cells with "heavier" species such as [$^{13}C_6$]-inositol. However, standard SAX-HPLC using water–salt-based gradients is incompatible with MS detection and MS-compatible volatile buffers do not currently enable isomer assignment[15]. HPLC-MS/MS and hydrophilic interaction liquid chromatography coupled to MS (HILIC-MS/MS) unfortunately result in a suboptimal separation of the analytes, obliterating InsP isomer identity[16].

The development of resolving methods using an electric field to separate the differentially charged InsPs has been pursued. High-voltage paper chromatography[17] was instrumental in the discovery and establishment of Ins(1,4,5)P$_3$ as the Ca$^{2+}$ release factor[18]. The separation of higher InsPs by gel electrophoresis (PAGE) is another possibility; it, however, does not have the resolving power to distinguish PP-InsP regioisomers and does not detect lower InsPs due to staining inefficiency[19].

A capillary electrophoresis (CE) MS method is described herein, that complements and significantly improves analytical approaches in the field. It does not require derivatization, benefits from the separation efficiency of charged analytes by CE, and enables accurate isomer identification and quantitation using stable-isotope-labeled (SIL) reference compounds, even in complex matrices. This setup also enables stable isotope pulse labeling experiments to analyze the amount of endogenously synthesized inositol over time.

## Results

**Development of CE-ESI-MS for the analysis of InsPs and PP-InsPs.** CE is known as an effective separation tool for phosphate-containing molecules. An early attempt to implement CE was made in the study of Ins(1,4,5)P$_3$, but the method was not developed further[20]. We now introduce a capillary electrophoresis coupled to electrospray ionization mass spectrometry (CE-ESI-MS) method, using a bare fused silica capillary and a simple background electrolyte (BGE), for parallel analyses of PP-InsPs and InsPs.

A set of PP-InsP and InsP standards (Fig. 1b), representing mammalian metabolites (Fig. 1a and Supplementary Fig. 1), including Ins(1,4,5)P$_3$, Ins(1,3,4,6)P$_4$, Ins(1,4,5,6)P$_4$, Ins(1,3,4,5)P$_4$, Ins(2,3,4,5,6)P$_5$, Ins(1,3,4,5,6)P$_5$, Ins(1,2,3,4,6)P$_5$, Ins(1,2,3,4,5)P$_5$, InsP$_6$, 5-PP-InsP$_5$, 1-PP-InsP$_5$, and 1,5-(PP)$_2$-InsP$_4$, was resolved with a BGE (35 mM ammonium acetate adjusted to pH 9.7 with NH$_4$OH) by applying a 30 kV voltage over a regular bare fused silica capillary with a length of 100 cm. Detection of analytes was achieved with an ESI-TOF-MS instrument in the negative ionization mode. A stable CE separation current (23 µA) and ESI spray current (2.1 µA) were maintained with a sheath flow CE-ESI-MS interface. The limits of quantitation (LOQs) for different InsPs were 150–500 nM (Supplementary Fig. 2), with 10 nL sample injection volume, i.e. 1.5–5.0 fmol of analyte. As baseline separation for the analytes is achieved, no issues with the inevitable in-source fragmentation with neutral loss of phosphate are encountered for accurate quantitation. Generally, <10% in-source fragmentation products were produced from doubly charged anionic forms of InsP$_5$ to InsP$_8$ with neutral loss of phosphate (79.97 Da).

Next, we validated the CE-ESI-MS protocol in analyzing InsPs from biological samples. Initially, we focused on two HCT116 cell lines (HCT116$^{UCL}$ and HCT116$^{NIH}$) that have been shown to possess different PP-InsP levels[21]. The CE-ESI-MS method was fully compatible with current state-of-the-art InsP extraction by perchloric acid followed by enrichment with TiO$_2$ (Fig. 1c). We introduced and resolved in parallel SIL internal standards (IS) of [$^{13}C_6$]1,5-(PP)$_2$-InsP$_4$, [$^{13}C_6$]5-PP-InsP$_5$, [$^{13}C_6$]1-PP-InsP$_5$, [$^{13}C_6$]InsP$_6$, and [$^{13}C_6$]Ins(1,3,4,5,6)P$_5$ (ref. [22]). Application of SIL standards is crucial, as the assignment of InsPs, particularly regioisomers, now becomes unambiguous. Spiking with precise amounts of SIL standards into a biological extract also enables a reliable quantitative assessment, since they compensate for matrix effects and analyte loss. Ins(1,3,4,5,6)P$_5$, InsP$_6$, and 5-PP-InsP$_5$ were assigned by their isotopic pattern, accurate mass, and identical migration time with spiked SIL standards. Excellent resolution and column efficiency were obtained: $1.5 \times 10^4$, $4.6 \times 10^4$, $3.0 \times 10^4$ theoretical plates per meter for 5-PP-InsP$_5$, InsP$_6$, and Ins(1,3,4,5,6)P$_5$, isolated from HCT116$^{NIH}$ extract, respectively (Fig. 1c). Analysis of HCT116$^{UCL}$ extract found 1,5-(PP)$_2$-InsP$_4$ (Fig. 2aI), a signal which was generally under the LOQ (250 nM) but within limit of detection (LOD) in HCT116$^{NIH}$

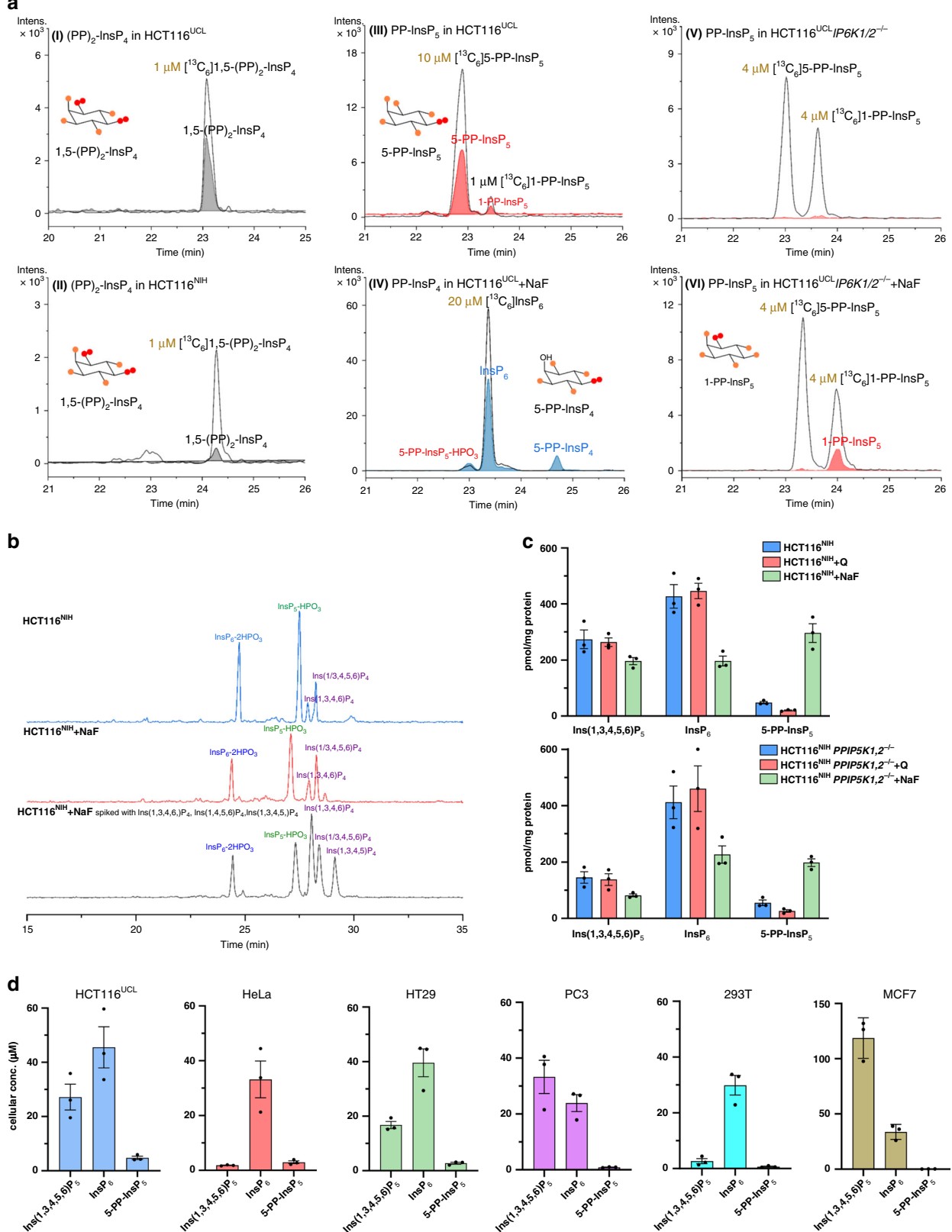

extract (Fig. 2aII), consistent with previous observations[21]. Furthermore, the analysis of HCT116[UCL] confirmed the presence of the less abundant 1-PP-InsP_5 isomer, representing <10% of the cellular PP-InsP pool (Fig. 2aIII). This is an important step forward for characterizing PP-InsP metabolism using MS. The TiO_2 extraction method was previously reported

to fully recover PP-InsPs and InsP_6 from mammalian cell extracts[23]. We confirmed this observation by spiking the HCT116[NIH] samples with SIL IS both before extraction (pre-spiking) and before measurement (post-spiking) (Supplementary Fig. 3). A flow chart depicting the established CE-MS procedure is included in Supplementary Fig. 4.

**Fig. 2 CE-ESI-MS analysis of mammalian PP-InsP and InsP metabolism. a** Mammalian cell extracts were analyzed by CE-ESI-MS after spiking the indicated amount (yellow) of stable isotopic standard. Extracted ion electropherograms (EIE) of 1,5-(PP)$_2$-InsP$_4$ in HCT116$^{UCL}$ (I) and HCT116$^{NIH}$ (II) spiked with [$^{13}$C$_6$] 1,5-(PP)$_2$-InsP$_4$, (III) PP-InsP$_5$ in HCT116$^{UCL}$, major 5-PP-InsP$_5$ and minor 1-PP-InsP$_5$, (IV) 5-PP-InsP$_4$ in HCT116$^{UCL}$ cells with NaF treatment, (V) no measurable PP-InsP$_5$ in HCT116$^{UCL}$IP6K1,2$^{-/-}$, (VI) 1-PP-InsP$_5$ is detectable in HCT116$^{UCL}$IP6K1,2$^{-/-}$ cells after 1 h of sodium fluoride (NaF) treatment. **b** Analysis of InsP$_4$ isomers in HCT116$^{NIH}$, HCT116$^{NIH,NaF}$, and HCT116$^{NIH,NaF}$ spiked with Ins(1,3,4,6)P$_4$ (2 μg/mL), Ins(1,4,5,6)P$_4$ (1 μg/mL), and Ins(1,3,4,5)P$_4$ (2 μg/mL). **c** Inositol (pyro)phosphate levels in HCT116$^{NIH}$ and HCT116$^{NIH}$PPIP5K1,2$^{-/-}$ cells, without treatment or after incubation with NaF or inositol polyphosphate kinase (IPMK) inhibitor quercetin (2.5 μM for 30 min). Bars are means ± SEM from three independent experiments, individual values are shown by dots. **d** Cellular concentration of InsPs in the immortal human cell lines HCT116$^{UCL}$, HeLa, HT29, 293T, PC3, and MCF7. Concentrations varying between 2–120 μM Ins(1,3,4,5,6)P$_5$, 24–46 μM InsP$_6$, and 0–5 μM 5-PP-InsP$_5$ were found. Data are means ± SEM from three independent experiments. InsP levels normalized by protein content among these cells are shown in Supplementary Fig. 6. Source data are provided as a Source Data file.

The CE-ESI-MS protocols are not limited to analysis of InsP$_6$ and PP-InsPs. Using the described conditions, Ins(1/3,2,4,5,6)P$_5$ and Ins(1,2,3,4,6)P$_5$ or Ins(1,2,3,4/6,5)P$_5$ were readily distinguished from Ins(1,3,4,5,6)P$_5$ (Supplementary Fig. 5a). Similarly, identification of InsP$_4$ positional isomers was achieved by measuring the accurate mass in combination with spiking of InsP$_4$ standards Ins(1,3,4,6)P$_4$, Ins(1,4,5,6)P$_4$, and Ins(1,3,4,5)P$_4$ (Fig. 2b).

Owing to minimal sample consumption (10 nL) and rapid analysis time (30 min) per run, measurement of technical replicates is feasible. The intra- and interday repeatabilities of the method analyzing HCT116$^{NIH}$ extract were evaluated, with mean relative standard deviations (RSDs) of 3.0% (Day 1), 3.1% (Day 2), and 6.3% (Day 3) for three technical replicates, and a mean RSD of 3.6% for technical replicates from five individual days (Fig. 1d). A comparison with the repeatabiliy of SAX-HPLC studies is difficult: details of technical replicates are often not provided, since the analysis of one sample takes about 6–8 h with hands-on processing[24,25].

**CE-ESI-MS analysis of mammalian PP-InsP metabolism.** The CE-ESI-MS method was also applied to monitor changes in PP-InsPs metabolism in mammalian cells in which their synthetic enzymes have been knocked out or perturbed using inhibitors. In mammals, PP-InsPs are synthesized by two different classes of enzymes (Supplementary Fig. 1). The IP6Ks, by phosphorylating position 5 of InsP$_6$, synthesize 5-PP-InsP$_5$. The PPIP5Ks are bifunctional (kinase/phosphatase) enzymes that, by acting on position 1, mainly convert 5-PP-InsP$_5$ into 1,5-(PP)$_2$-InsP$_4$. CE-ESI-MS analysis of HCT116$^{UCL}$IP6K1,2$^{-/-}$ confirms prior observations that PP-InsPs are absent (Fig. 2aV)[26]. Levels of PP-InsPs can be increased by blocking their dephosphorylation using sodium fluoride (NaF, 10 mM)[27]. CE-ESI-MS analysis of NaF-treated HCT116$^{UCL}$ (Supplementary Fig. 5) or HCT116$^{NIH}$ cells (Fig. 2c) demonstrated the expected 5-PP-InsP$_5$ elevation (7.3-fold in HCT116$^{UCL}$ cells and 6.2-fold in HCT116$^{NIH}$ cells) with concomitant reduction in InsP$_6$. We observed a reduction in Ins(1,3,4,5,6)P$_5$ levels and appearance of 5-PP-InsP$_4$ (Fig. 2aIV), changes also observable by SAX-HPLC analysis of [$^3$H]-inositol-labeled HCT116$^{UCL}$ (Supplementary Fig. 5). The synthesis of 5-PP-InsP$_4$ is dependent on IP6Ks acting on Ins(1,3,4,5,6)P$_5$: consistent with this, CE-ESI-MS analysis of NaF-treated HCT116$^{UCL}$IP6K1,2$^{-/-}$ also failed to detect any 5-PP-InsP$_4$. However, confirming a previous observation[26], 1-PP-InsP$_5$ became detectable in HCT116$^{UCL}$IP6K1,2$^{-/-}$ after NaF treatment (Fig. 2aVI). This is explained by PPIP5Ks' capacity to use InsP$_6$ as a substrate, particularly when its preferred substrate, 5-PP-InsP$_5$, is absent. We also observed an increase in Ins(1/3,4,5,6)P$_4$ levels in NaF-treated HCT116$^{NIH}$ cells (Fig. 2b) as a result of PLC activation[27]. Analysis of HCT116$^{NIH}$PPIP5K1,2$^{-/-}$ in comparison to wild-type cells showed a small increase of the non-metabolized

substrate 5-PP-InsP$_5$, the levels of which are 3.6-fold enhanced by NaF treatment (Fig. 2c). We additionally analyzed the effect of a recently identified IP6K inhibitor: quercetin (Q)[28] reduced 5-PP-InsP$_5$ levels in both HCT116$^{NIH}$ and HCT116$^{NIH}$PPIP5K1,2$^{-/-}$ cells by 50–60% (Fig. 2c).

These results validate CE-ESI-MS as a technique to accurately monitor changes in cellular InsPs and PP-InsPs metabolism in response to different stressors or genetic alterations.

**Analysis of InsPs in mammalian cell lines and tissues**. We determined the concentrations of Ins(1,3,4,5,6)P$_5$, InsP$_6$, and 5-PP-InsP$_5$, in different mammalian cell lines, including HCT116$^{UCL}$, HeLa, HT29, PC3, 293T, and MCF7 (Fig. 2d, Supplementary Fig. 6). The detected InsP$_6$ and 5-PP-InsP$_5$ cellular concentrations as well as their relative ratio are in accordance with earlier results obtained by [$^{13}$C]-NMR[29]. The InsP$_6$ cellular concentration, for example, was in the range of 24–47 μM (300–500 pmol/mg protein). However, Ins(1,3,4,5,6)P$_5$ levels were surprisingly variable, potentially reflecting different functional roles (Supplementary Fig. 7) across different cell lines. Therefore, the CE-MS method will be instrumental to uncover dynamics and physiological roles of InsP$_5$ in mammalian cells.

We quantified the amount of InsPs and PP-InsPs in mouse organs, including liver, brain, muscle, kidney, and spleen (Supplementary Fig. 8). Again, Ins(1,3,4,5,6)P$_5$, InsP$_6$, and 5-PP-InsP$_5$ were the main InsP species. Comparably low InsP levels were detected in muscle. Analysis of InsP$_6$ and 5-PP-InsP$_5$ extracted from tissues performed by PAGE lack the sensitivity, resolution, and dynamic range, evidenced by CE-ESI-MS. The analysis of InsPs from animal organs or tissues cannot be performed by SAX-HPLC, prohibited by cost, feasibility, and ethical considerations, since it requires feeding a mouse with [$^3$H]-inositol.

**InsPs and PP-InsPs in *Saccharomyces cerevisiae* and *Arabidopsis thaliana*.** We next analyzed InsPs and PP-InsPs in yeast and plant samples to explore the applicability of CE-ESI-MS across experimental models. The [$^3$H]-inositol SAX-HPLC method has been extensively employed to study yeast and plant InsP metabolism. PAGE methods conversely cannot be used to analyze InsPs from yeast, due to the abundant inorganic polyphosphate (polyP) that suppresses the InsP signals (Supplementary Fig. 9a), and while PAGE has been applied to study InsPs including PP-InsPs in plant extracts[4,30,31], the same limitations described above apply. Using SIL-CE-ESI-MS, profiling of InsPs and PP-InsPs was readily achieved for both *Saccharomyces cerevisiae* and *Arabidopsis thaliana*.

In wild-type yeast extracts, InsP$_6$, 5-PP-InsP$_5$, and 1/3,5-(PP)$_2$-InsP$_4$ were detectable. In agreement with the literature[32], 5-PP-InsP$_5$ and 1/3,5-(PP)$_2$-InsP$_4$ were around 3% and 1% of the InsP$_6$ level, respectively (Supplementary Fig. 9).

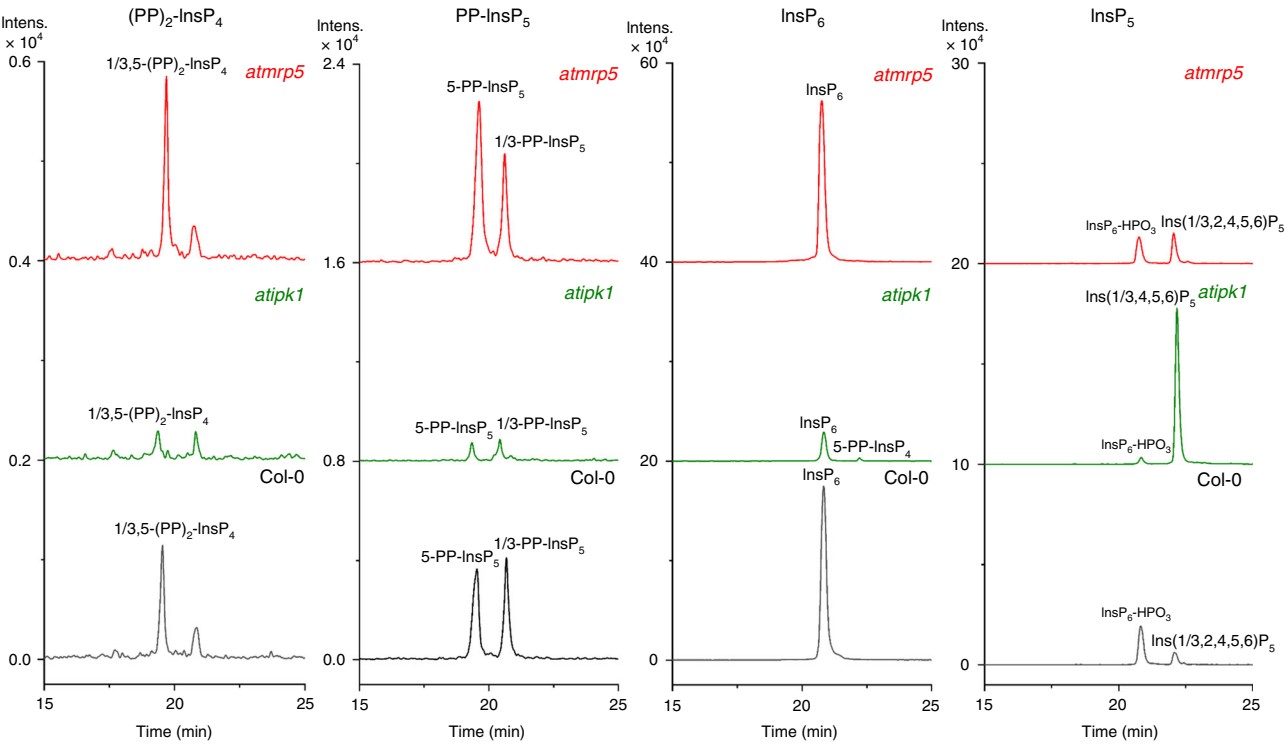

**Fig. 3 CE-ESI-MS analysis of PP-InsP in *Arabidopsis thaliana* shoots.** Shoot extracts from wild type (Col-0), *atipk1*, and *atmrp5* mutant plants were analyzed to assess the presence of InsP$_5$, InsP$_6$, 5-PP-InsP$_5$, and 1,5-(PP)$_2$-InsP$_4$. InsP$_5$ isomer composition differed in the different genotypes: Ins(1/3,2,4,5,6)P$_5$ is present in Col-0 and the *mrp5* mutant, whereas Ins(1,3,4,5,6)P$_5$ was detected in *atipk1* plants. A decreased level of 1,5-(PP)$_2$-InsP$_4$, 5-PP-InsP$_5$, 1-PP-InsP$_5$, InsP$_6$, and increased level of 5-PP-InsP$_4$ and Ins(1,3,4,5,6)P$_5$ were observed for the *atipk1* mutant. In the *atmrp5* mutant, levels of 1,5-(PP)$_2$-InsP$_4$ and 5-PP-InsP$_4$ were increased. The electropherograms are representative of independent biological duplicates giving comparable results. Source data are provided as a Source Data file.

We analyzed the InsPs present in shoots of *A. thaliana* wild type (Col-0) and in plants defective in inositol pentakisphosphate 2-kinase (AtIPK1) or the ATP-binding cassette (ABC) transporter 5 (AtMRP5)[33] that transports InsP$_6$ into the vacuole (Fig. 3): Ins(1/3,2,4,5,6)P$_5$, InsP$_6$, 5-PP-InsP$_5$, 1/3-PP-InsP$_5$, and 1/3,5-(PP)$_2$-InsP$_4$ were readily detected in the shoot extracts of *A. thaliana* Col-0 seedlings. Surprisingly, comparable levels of 5-PP-InsP$_5$ and 1/3-PP-InsP$_5$ were observed. This represents a significant deviation from the general notion, derived from mammalian cell analysis, that 1-PP-InsP$_5$ represents the minor isomeric species. The *atipk1* mutant displayed decreased levels of PP-InsPs and InsP$_6$, and a robust increase in Ins(1,3,4,5,6)P$_5$ s. In *atipk1* plants, 5-PP-InsP$_4$ was also detected by CE-ESI-MS, which was previously unsuccessfully tracked by [³H]-inositol labeling and SAX-HPLC analysis[34]. Analysis of *atmrp5* shoots revealed the expected elevated levels of 5-PP-InsP$_5$ and 1/3,5-(PP)$_2$-InsP$_4$, but not of 1/3-PP-InsP$_5$. This analysis provides insights into single isomer PP-InsP alterations in plants, underlining the value of the SIL-CE-ESI-MS method for dissecting PP-InsP isomers and relative abundances.

**CE-ESI-MS to analyze *D. discoideum* PP-InsP metabolism.** The social amoeba *D. discoideum* possesses large amounts of PP-InsPs. However, this model organism contains different PP-InsPs isomers from those in mammals, yeast, and plants, such as 6-PP-InsP$_5$ and 5,6-(PP)$_2$-InsP$_4$ (refs. [10,35]). This complexity is a challenge for ideal CE separation conditions. Employing a different BGE with decreased ionic strength and pH (30 mM ammonium acetate adjusted to pH 9.0 with NH$_4$OH) led to enhanced resolution of InsP$_6$ and more anionic species. Under these conditions, all possible PP-InsP$_5$ isomers, including non-

natural ones obtained by chemical synthesis[36–38], were separated and could thus be assigned in complex matrices (Fig. 4a). This BGE also enabled the separation of all available InsP$_5$ isomers. We then performed CE-ESI-MS analyses of *D. discoideum* extracts (Fig. 4b). As expected, 4/6-PP-InsP$_5$ was about twice as abundant as 5-PP-InsP$_5$, while 1/3-PP-InsP$_5$ was present at around 5% of the whole PP-InsP$_5$ pool. We identified two (PP)$_2$-InsP$_4$ isomers, 5,4/6-(PP)$_2$-InsP$_4$, and 1,5-(PP)$_2$-InsP$_4$. Peak assignment and quantitation were conducted in a single run (30 min), and accurate mass information was provided simultaneously (Fig. 4c).

**Endogenous inositol synthesis contributes to the InsP pools.** The ability of MS to capture isotopic mass differences is a significant advantage of the CE-ESI-MS protocol. For example, by using isotopically labeled metabolic precursors, the contribution to the InsP pool of both the inositol acquired from the milieu and from endogenous synthesis from glucose can now be assessed (Fig. 1a). In SAX-HPLC analysis, inositol-free medium is used to improve [³H]-inositol labeling, with the tracer used at 0.5–1 μM concentration[25]. This method does not detect endogenously generated inositol. To assess the contribution of endogenous synthesis of inositol to InsP cellular pools, we performed an inositol titration curve. Wild-type HCT116$^{UCL}$ were incubated for 5 days, the incubation time used for [³H]-inositol labeling to reach metabolic equilibrium (7–8 cell division cycles), in inositol-free DMEM in the presence of 1, 10, and 100 μM [¹³C$_6$]-inositol. We observed a dose-dependent incorporation of [¹³C$_6$]-inositol into the [¹³C$_6$]InsP$_5$ and [¹³C$_6$]InsP$_6$ pools (Fig. 5a). Using 1 μM [¹³C$_6$]-inositol, <20% of the InsP$_5$ and InsP$_6$ pools were synthesized from exogenously acquired inositol; this value increased to

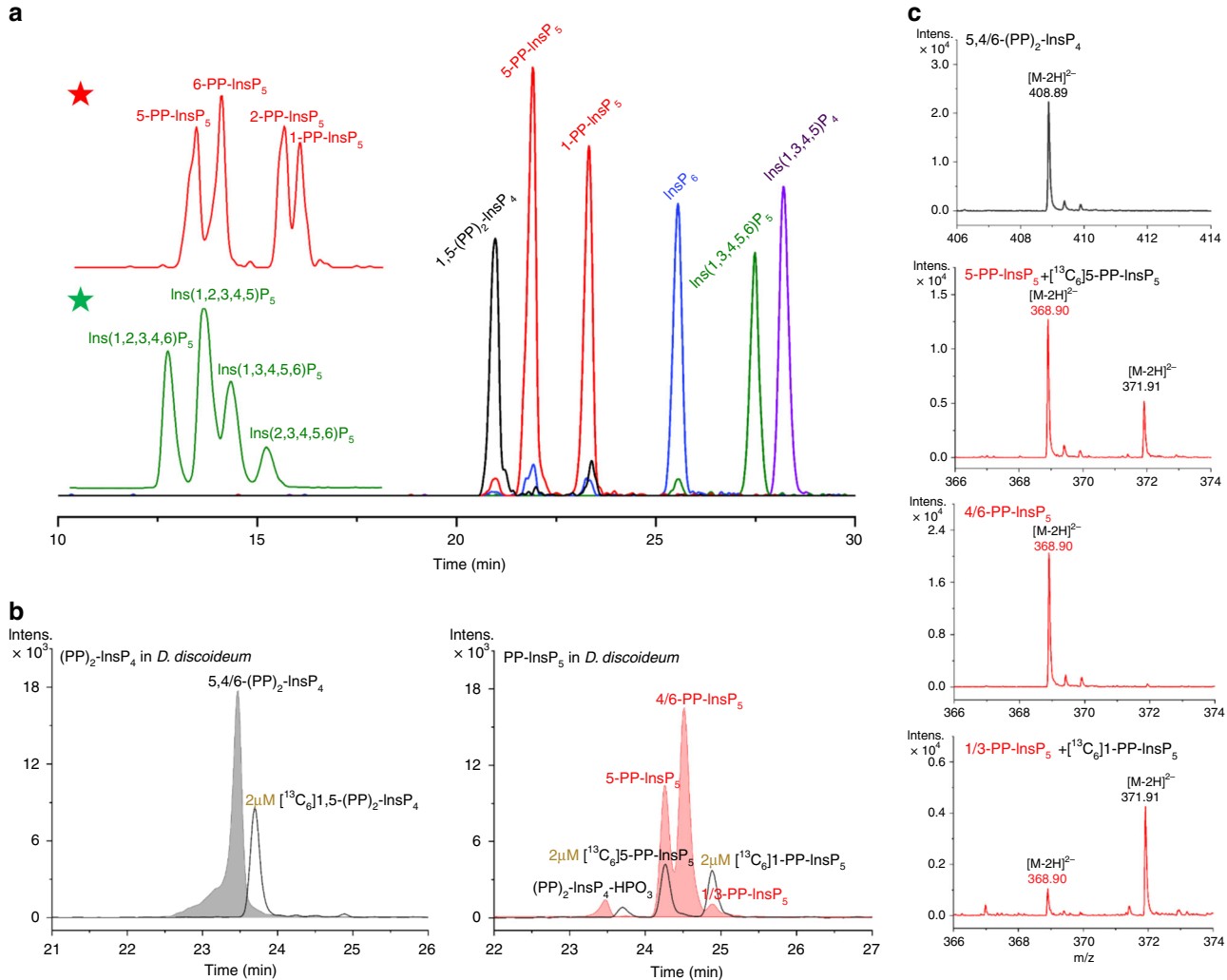

**Fig. 4 CE-ESI-MS analysis of *Dictyostelium discoideum* extract. a** Separation of PP-InsP and InsP standards using a modified BGE (30 mM ammonium acetate titrated with NH$_4$OH to pH 9.0). Enlarged inserts for PP-InsP$_5$ (red star) and InsP$_5$ (green star) reveal the separation of all regioisomers using the following parameters: CE voltage: 30 kV, CE current: 19 µA, injection: 50 mbar, 10 s (10 nL), solutes: 10 µM for Ins(1,3,4,5,6)P$_5$, InsP$_6$, 5-PP-InsP$_5$, 1-PP-InsP$_5$, and 1,5-(PP)$_2$-InsP$_4$; 20 µg/mL for Ins(1,3,4,5)P$_4$. **b** PP-InsPs in *D. discoideum* extracts. PP-InsP regioisomers could be efficiently assigned and quantified in single runs with spiked (isotopic) standards: major 5,4/6-(PP)$_2$-InsP$_4$, 4/6-PP-InsP$_5$, and 5-PP-InsP$_5$, minor 1/3-PP-InsP$_5$. **c** Mass spectra for PP-InsPs in *D. discoideum* in **b.** Three independent *D. discoideum* AX2 (wild type) extracts were analyzed giving identical results. Source data are provided as a Source Data file.

90% using 100 µM. A reference value for inositol concentration in human serum is 30 µM[39], available only to cells in direct contact with serum. We next performed a time course, incubating cells with [$^{13}$C$_6$]-inositol (10 µM) in inositol-free DMEM for 1, 3, and 5 days (Fig. 5b). Initially, all InsP$_5$ and InsP$_6$ species were constituted of [$^{12}$C]-inositol. By 24 h, [$^{13}$C$_6$]InsP$_5$ and [$^{13}$C$_6$]InsP$_6$ became detectable, representing a small fraction of their respective pools. After 5 days, [$^{13}$C$_6$]InsP$_5$ and [$^{13}$C$_6$]InsP$_6$ represented two-thirds of the InsP$_5$ and InsP$_6$ pools. These results indicate either a sluggish InsP$_5$ and InsP$_6$ turnover, even more lethargic than previously thought[40], or an endogenous synthesis of [$^{12}$C]-inositol that substantially contributes to the InsP$_5$ and InsP$_6$ pools.

The inositol phosphate synthase (IPS or MIPS) called ISYNA1 in mammals converts glucose-6-phosphate to Ins(3)P[41]. To study the contribution of endogenous inositol synthesis to InsP pools, we generated two independent HCT116$^{UCL}$ *ISYNA1*$^{-/-}$ clones using CRISPR. The two knockout clones, KO1 and KO2, were verified by sequencing (Supplementary Fig. 10) and western blot analysis (Fig. 6a). The InsP$_6$ level as analyzed by PAGE was

unaffected in the *ISYNA1*$^{-/-}$ clones (Fig. 6b), and there was no growth defect in normal medium (Fig. 6c). Titrating the medium with different inositol concentrations revealed 10 µM inositol as sufficient to guarantee wild-type growth rate, while the absence of inositol dramatically reduced the growth of both clones (Fig. 6d–f). To study the contribution of ISYNA1 to the InsP pools, we incubated wild-type HCT116$^{UCL}$ and *ISYNA1*$^{-/-}$ clones for 5 days in 25 mM [$^{13}$C$_6$]-glucose with 1 or 10 µM inositol, and then extracted and analyzed the samples by CE-ESI-MS (Fig. 6g). In wild-type cells with only 1 µM inositol, roughly 60% of the InsP$_5$ and InsP$_6$ pools were generated by the endogenous conversion of [$^{13}$C$_6$]-glucose-6-phosphate to Ins(3)P, detected as [$^{13}$C$_6$]InsP$_5$ and [$^{13}$C$_6$]InsP$_6$. Note that often 1 µM or less of inositol is used in [$^3$H]-inositol labeling for SAX-HPLC experiments[26,42]. Our analysis, therefore, reveals that up to 60% of InsPs are not detected by traditional methods if inositol concentration is kept low to improve [$^3$H]-inositol labeling efficiency. In the presence of a tenfold higher exogenous inositol concentration (10 µM), endogenously generated inositol contributed 15% to the InsP$_5$ and InsP$_6$ pools. Strikingly, we detected

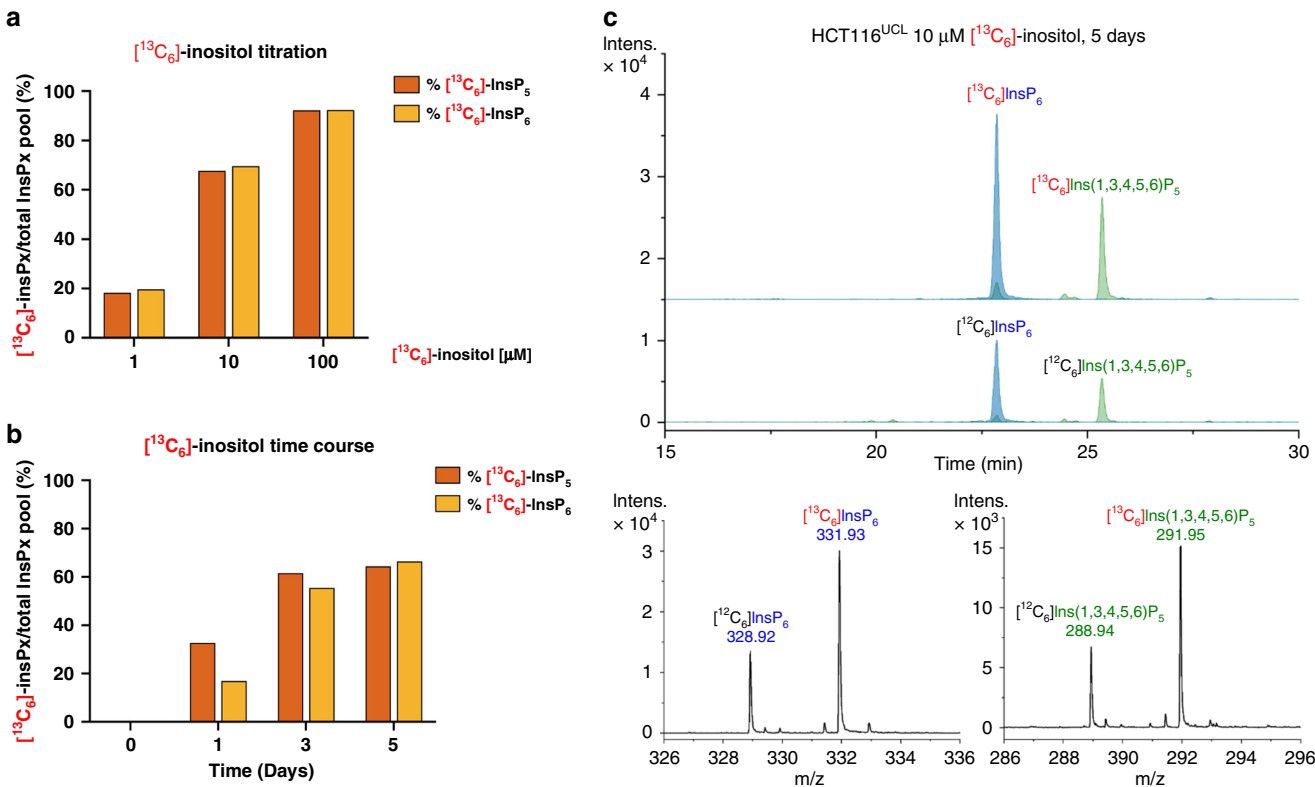

**Fig. 5 CE-ESI-MS analysis after [$^{13}$C$_6$]-inositol labeling. a** Ratio of [$^{13}$C$_6$]InsP$_x$ ($x = 5$ or 6) over total InsP$_x$ ([$^{13}$C$_6$]InsP$_x$ + [$^{12}$C$_6$]InsP$_x$) pool in wild-type HCT116$^{UCL}$ after 5 days incubation with different concentrations of [$^{13}$C$_6$]-inositol (1, 10, 100 µM) in inositol-free DMEM. **b** Ratio of [$^{13}$C$_6$]-InsP$_x$ in total InsP$_x$ pool in wild-type HCT116$^{UCL}$ cells incubated with 10 µM [$^{13}$C$_6$]-inositol in inositol-free DMEM for 1, 3, and 5 days. **c** EIEs of InsP$_6$ and Ins(1,3,4,5,6)P$_5$ in HCT116$^{UCL}$ cells after 5 days incubation with 10 µM [$^{13}$C$_6$]-inositol and relative mass spectra. Source data are provided as a Source Data file.

[$^{13}$C$_6$]InsP$_5$ and [$^{13}$C$_6$]InsP$_6$ in both *ISYNA1*$^{-/-}$KO1 and *ISYNA1*$^{-/-}$KO2, although to a lesser extent than in wild-type cells (Fig. 6h–j). This unexpected result indicates the existence of an alternative uncharacterized enzymology for inositol synthesis, underscoring the potential of the CE-ESI-MS technique. This approach will be instrumental not only in elucidating the inositol synthetic route we have discovered, but also in any future assessment of inositol phosphate physiological functions across the kingdoms of life.

## Discussion

The analysis of InsP and PP-InsP turnover in cells and tissues is a daunting challenge. Radiolabeling followed by SAX-HPLC has so far been the method of choice, as it provides the sensitivity needed for meaningful analyses of the less abundant species, with the advantage of selectively visualizing only InsPs and PP-InsPs when [$^3$H]-labeled inositol is fed to cells. This approach, however, misses inositol endogenously synthesized from glucose, is restricted to specialized laboratories, is expensive, and time consuming. While other approaches, such as [$^{13}$C] labeling for NMR, PAGE, and HILIC-MS, have recently been developed to provide alternatives, a transformative approach is still missing.

Here, we have demonstrated that CE is an effective separation platform for InsPs and PP-InsPs. Moreover, CE coupling to an ESI-Q-TOF mass spectrometer facilitates parallel analyses of a multitude of analytes in a single run, requiring only 30 min and nanoliter sample injection. According to the accurate mass information and identical mobility with (isotopic) standards, these densely charged species can now be readily assigned even in complex matrices. Additionally, the introduction of SIL reference compounds allows for quantitation and correction of matrix effects in samples such as

those obtained from yeast extracts rich in polyphosphates, where drifts in migration times of several minutes were observed. Using this approach, we were able to quantitate InsPs and PP-InsPs in different species, and in wild-type and knock-out cell lines additionally treated with inhibitors of several enzymes. Given the rapidity of the analysis, measurement of technical replicates becomes possible, underlining the robustness and fidelity of the method. Using CE-ESI-MS, we have been able to extract essential new information from several samples. For example, Ins(1,3,4,5,6)P$_5$ concentrations are highly variable across different mammalian cell lines and whole organs. We also show that 1/3-PP-InsP$_5$ is not always the minor isomer present, as exemplified by our analysis of *A. thaliana* seedlings, raising questions concerning its potential regulatory effects. Moreover, SIL inositol and SIL D-glucose were used in pulse labeling experiments, demonstrating that SIL-CE-ESI-MS can be used to monitor and quantitate inositol isomer turnover originating from different sources. At low (1 µM) exogenous inositol concentration, around 60% of cellular inositol was derived from glucose after 5 days of labeling. Yet, knockout of the only known glucose-6-phosphate utilizing inositol synthase in mammalian cells (ISYNA1) did not lead to cells incompetent in transforming D-glucose to inositol. This finding reveals that there must exist a yet uncharacterized biochemical pathway for the synthesis of inositol deriving its carbon skeleton from glucose. We thus conclude that SIL-CE-ESI-MS will open our eyes to cellular pathways we have previously been blind to.

## Methods

**Materials and reagents.** InsP$_6$, Ins(1,3,4,5,6)P$_5$, Ins(2,3,4,5,6)P$_5$, Ins(1,2,3,4,6)P$_5$, Ins(1,2,3,4,5)P$_5$, Ins(1,3,4,5)P$_4$, and Ins(1,4,5)P$_3$ with purity more than 95–97% ([$^{31}$P] NMR) were purchased from Sichem. Ins(1,3,4,6)P$_4$ and Ins(1,4,5,6)P$_4$ were obtained from Cayman. 1-PP-InsP$_5$, 5-PP-InsP$_5$, 6-PP-InsP$_5$, and 2-PP-InsP$_5$ were

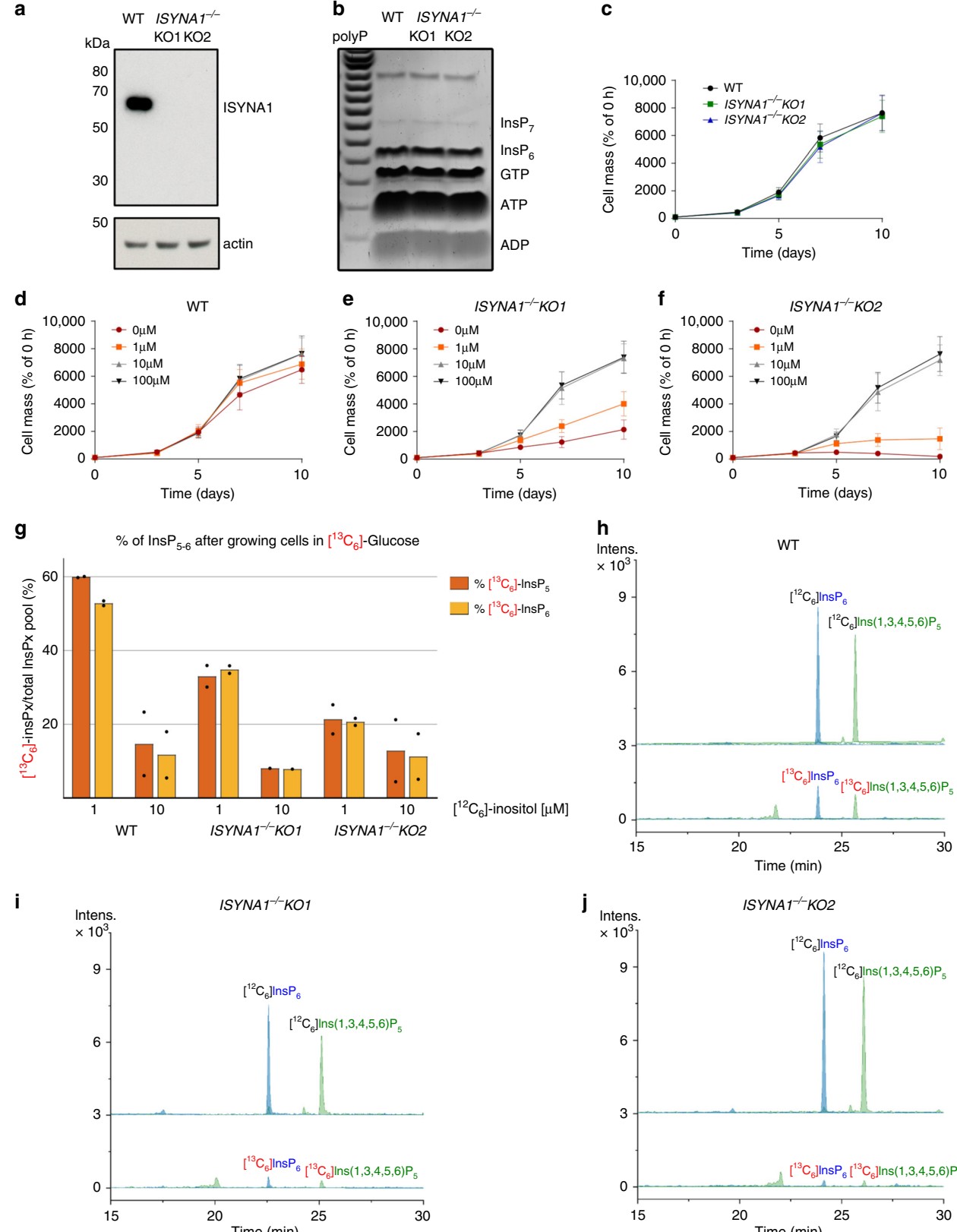

synthesized in-house (Jessen)[36–38]. SIL internal standards (IS) [$^{13}C_6$]1,5-(PP)$_2$-InsP$_4$, [$^{13}C_6$]5-PP-InsP$_5$, [$^{13}C_6$]1-PP-InsP$_5$, [$^{13}C_6$] InsP$_6$, and [$^{13}C_6$]Ins(1,3,4,5,6)P$_5$ with purities higher than 96% were synthesized in-house (Fiedler)[22,29]. Concentrations of stock solutions of InsP and PP-InsP standards for quantitation were determined by [$^1$H] NMR and/or [$^{31}$P] NMR as described below. Fused silica capillaries were obtained from CS-Chromatographie.

**CE-ESI-MS analysis.** All experiments were performed on a bare-fused silica capillary with a length of 100 cm (50 μm internal diameter and 365 μm outer diameter) on an Agilent 7100 capillary electrophoresis system coupled to a Q-TOF (6520, Agilent) equipped with a commercial CE-MS adapter and sprayer kit from Agilent. Data were collected with Agilent OpenLAB CDS Chemstation 2.3.53 and Agilent MassHunter Workstation Acquisition for Q-TOF B.04.00.

**Fig. 6 Analysis of *ISYNA1*$^{−/−}$ cell lines by CE-ESI-MS. a** Western blot showing the absence of ISYNA1 protein in CRISPR-generated HCT116$^{UCL}$ *ISYNA1* $^{−/−}$ clones KO1 and KO2, the result is representative of three independent analyses. These lines possess a normal level of InsP$_6$ as revealed by PAGE (**b**), the result is representative of experiments run three times, independently. The polyP loaded lane was used to define sample order, and a normal growth rate in standard DMEM (**c**). The results are means ± SD from three independent experiments performed in triplicate. Inositol titration growth curves (**d-f**) demonstrated strongly reduced growth for the KO cells in the absence of inositol, while a concentration of 10 µM inositol is sufficient to restore normal growth. The results are means ± SD from three independent experiments performed in triplicate. **g** Ratio of [$^{13}$C$_6$]InsP$_x$ over total InsP$_x$ pool ($x$ = 5 or 6) in WT, KO1, and KO2 clones grown for 5 days in 25 mM [$^{13}$C$_6$]-glucose supplemented with 1 or 10 µM inositol. Data are means from two independent experiments. **h** EIEs of InsP$_6$ and Ins(1,3,4,5,6)P$_5$ in HCT116$^{UCL}$ cells after 5 days in 25 mM [$^{13}$C$_6$]-glucose and 10 µM inositol, supporting a pathway from glucose for the biosynthesis of inositol phosphates in mammals. **i, j** EIEs of InsP$_6$ and Ins(1,3,4,5,6)P$_5$ in KO1 and KO2 cells after 5 days in 25 mM [$^{13}$C$_6$]-glucose with 10 µM inositol. Detectable [$^{13}$C$_6$]InsP$_x$ was produced in both clones although to a lesser degree than in wild type (**h**). Source data are provided as a Source Data file.

Prior to use, the capillary was flushed for 10 min with 1 N NaOH, followed by water for 10 min, and BGE for 15 min. BGE A (35 mM ammonium acetate titrated by ammonia solution to pH 9.7) was employed for the analysis of all mammalian cell and tissue extracts, as well as *S. cerevisiae* and *A. thaliana* extracts. BGE B (30 mM ammonium acetate titrated by ammonia solution to pH 9.0) was used for *D. discoideum*. Samples were injected by applying 50 mbar pressure for 10 s, corresponding to 0.5% of the total capillary volume (10 nL). In the study of the endogenous inositol synthesis, samples were injected with 100 mbar pressure for 10 s (20 nL). After sample injection, a BGE post-plug was introduced by applying 50 mbar for 2 s. For each analysis, a constant CE current of either 23 µA (BGE A) or 19 µA (BGE B) was established by applying 30 kV over the capillary, which was kept at a constant temperature of 25 °C.

The sheath liquid was composed of a water–isopropanol (1:1) mixture spiked with mass references. It was introduced at a constant flow rate of 1.5 µL/min. ESI-TOF-MS was conducted in the negative ionization mode; the capillary voltage was set to −3000 V and stable ESI spray current at 2.1 µA. For TOF-MS, the fragmentor, skimmer, and Oct RFV voltage was set to 140, 60, and 750 V, respectively. The temperature and flow rate of drying gas was 250 °C and 8 L/min, respectively. Nebulizer gas pressure was 8 psi. Automatic recalibration of each acquired spectrum was performed using reference masses of reference standards (TFA anion, [M-H]$^−$, 112.9855), and (HP-0921, [M-H+CH$_3$COOH]$^−$, 980.0163). Exact mass data were acquired at a rate of 1.5 or 1 spectra/s over a 60−1000 $m/z$ range. Extracted ion electropherograms (EIEs) were created using a 10-p.p.m. mass tolerance window for theoretical masses corresponding to the targeted inositol pyrophosphates and inositol phosphates.

Peak assignment of 1/3,5-(PP)$_2$-InsP$_4$, 5-PP-InsP$_5$, 1/3-PP-InsP$_5$, InsP$_6$, and Ins(1,3,4,5,6)P$_5$ in biological samples was achieved by accurate mass, isotopic pattern, and identical migration time. Ins(1,3,4,6)P$_4$, Ins(1/3,4,5,6)P$_4$, and 4/6-PP-InsP$_5$ in biological samples were assigned by accurate mass and identical migration time with spiked standards. 5-PP-InsP$_4$ and 4/6,5-(PP)$_2$-InsP$_4$ in biological samples were assigned by accurate mass and based on previous research[10,26].

Quantitation of PP-InsP and InsP in mammalian cells was performed with known amounts of individual isotopic standards spiked into the samples. The amount of isotopic IS that had to be added was delineated from the concentration of the respective analyte in the sample. Ratio of analyte peak area (Area)$^{12C}$ / IS peak area (Area)$^{13C}$ is <5 to ensure a linear relationship. IS stock solutions of 125 µM [$^{13}$C$_6$]5-PP-InsP$_5$, 500 µM [$^{13}$C$_6$]InsP$_6$, and 500 µM [$^{13}$C$_6$]Ins(1,3,4,5,6)P$_5$ were used for the spiking experiments; 0.4 µL of IS stock solution was added into 10 µL samples before measurement. 5 µM [$^{13}$C$_6$]5-PP-InsP$_5$, 20 µM [$^{13}$C$_6$]InsP$_6$ and 20 µM [$^{13}$C$_6$]Ins(1,3,4,5,6)P$_5$ were the final concentrations inside samples. The calibration curve for each analyte was constructed at eight levels by regression of nominal analyte concentration against a ratio of analyte peak area (Area)$^{12C}$ / IS peak area (Area)$^{13C}$ (Supplementary Fig. 11). The calibration curves were linear and provided a coefficient of determination >0.997 over the investigated range of concentrations (0.25–25 µM for 5-PP-InsP$_5$, 1.0–100 µM for InsP$_6$ and Ins(1,3,4,5,6)P$_5$). For quantitation, two technical replicates were conducted for each sample.

Quantitation of InsP and PP-InsP in tissue extracts, *S. cerevisiae*, *A. thaliana*, and *D. discoideum* extracts was performed by comparing analyte peak areas with the respective peak areas of SIL IS with known concentrations. Concentrations of SIL IS solutions were determined by quantitative $^{31}$P and $^1$H NMR against a certified reference standard (phosphoacetic acid, *Trace*CERT® Merk 96708–1G).

**Maintenance and manipulation of mammalian cell lines.** HCT116$^{NIH}$ and HCT116$^{UCL}$ were originally acquired from American Type Culture Collection (HCT 116, catalog CCL-247) and defined by tandem DNA repeat sequences[21]. From these lines, HCT116$^{NIH}$ *PPIP5K*$^{−/−}$ cells[43] and HCT116$^{UCL}$ *IP6K1,2*$^{−/−}$ cells[44] were derived. HeLa (ATCC, CCL-2), HT29 (ATCC, HTB-38), HEK293T (ATCC, CRL-11268), PC3 (ATCC, CRL-1435), and MCF7 (ATCC, HTB-22) cells were kind gifts from departmental colleagues.

HCT116$^{NIH}$ and HCT116$^{NIH}$ *PPIP5K*$^{−/−}$ cells[43] were grown in DMEM (Gibco) supplemented with 4.5 g/L glucose and 10% heat inactivated FBS (Gibco). HCT116$^{UCL}$ (obtained from European Collection of Authenticated Cell Cultures

[ECACC]) and HCT116$^{UCL}$ *IP6K1,2*$^{−/−}$ cells[44] were cultured in DMEM supplemented with 10% FBS (Sigma) and 4.5 g/L glucose. All cells were grown in a humidified atmosphere with 5% CO$_2$. InsP levels were modulated by incubating the cells for 60 min with 10 mM NaF or for 30 min with 2.5 µM quercetin[28] prior to harvesting.

For inositol limitation or [$^{13}$C$_6$]-inositol labeling experiments, inositol-free DMEM (MP Biomedicals) with 10% dialyzed FBS (Sigma) was used. Normal inositol (Sigma) or [$^{13}$C$_6$]-inositol[29] were supplemented as appropriate. Cells were acclimatized to 10 µM inositol in inositol-free DMEM for 1 week before starting labeling experiments. For [$^{13}$C$_6$]-glucose (Sigma) labeling experiments, DMEM lacking both inositol and glucose (Thermo Fisher) was used, using 10% dialyzed FBS. Cells were washed twice in the relevant starvation medium before incubation.

**Mammalian cell growth assay.** To measure cell growth, the sulforhodamine B (SRB) assay was performed[45]. Cells were seeded into 96-well plates. After 24 h, the medium was removed, wells were washed, then 100 µL of treatment medium was added. At each timepoint, cells were fixed in 10% trichloroacetic acid. Fixed plates were stained with 0.05% SRB (Sigma) in 1% acetic acid, and fixed dye was solubilized in 10 mM Tris base before reading absorbance at 500 nm using a spectrophotometer.

To measure cell volume, 80–90% confluent cells were trypsinized and resuspended in growth medium. A Multisizer 4 (Beckman Coulter) machine was used, following the manufacturer's instruction.

**Generation of ISYNA1 KO cell lines.** The human colon carcinoma cell line HCT116$^{UCL}$ was used to generate knockouts as it is pseudo-diploid, and has easily detectable amounts of InsP$_6$ and InsP$_7$ (ref. [23]). The Alt-R CRISPR-Cas9 (Integrated DNA Technologies) system was used, with guide sequence 5′-CCAAUCGACUGCGUU-3′. CRISPR components were introduced into the cells using a Neon electroporator (Thermo Fisher) and cells plated into 96-well plates using limiting dilution. Colonies were screened by western blotting using anti-ISYNA1 antibody (Santa Cruz sc-271830). Positive knockout clones were further confirmed by Sanger sequencing-based analysis (Genewiz CRISPR Analysis Package).

**Purification of inositol phosphates by titanium dioxide pulldown.** Extraction of inositol phosphates was performed according to the literature[44]. Briefly, 80–90% confluent cells were extracted using 1 M perchloric acid as described below. Titanium dioxide beads (Titansphere TiO 5 µm; GL Sciences) were used to pull down inositol phosphates, which were eluted using 3% ammonium hydroxide. The ammonia was removed and the samples concentrated using a speedvac evaporator for 1–3 h at 40 °C or 60 °C. For InsPs analysis by PAGE, the extracts were normalized to protein concentration and resolved using 35% PAGE gels[19]. Inositol phosphates were visualized by Toluidine blue (Sigma) staining. A desktop scanner (Epson) was used to record the PAGE result.

**Preparing cell extracts for CE-ESI-MS.** *Mammalian cells*: Cells (8 million HCT116$^{UCL}$ cells, HCT116$^{UCL}$ *IP6K1,2*$^{−/−}$ cells, and HCT116$^{NIH}$ *PPIP5K*$^{−/−}$ cells; 6 million HCT116$^{NIH}$ cells) were seeded into 15-cm dishes and allowed to grow for 48 h (HCT116$^{UCL}$ and HCT116$^{UCL}$ *IP6K1,2*$^{−/−}$ cells) or 72 h (HCT116$^{NIH}$ and HCT116$^{NIH}$ *PPIP5K*$^{−/−}$ cells) until 80–90% confluent. To harvest, dishes were quickly washed twice with cold PBS, then incubated with 1–5 mL cold 1 M perchloric acid on ice for 10 min. Acidic extracts were then collected from the plates, and inositol phosphates and other small polar molecules extracted using titanium dioxide beads[23,44]. To determine protein concentrations, post-extraction dishes were washed twice in PBS and proteins were solubilized via addition of 1.5 mL cell lysis buffer (0.1% SDS in 0.1 M NaOH) followed by incubation for 15 min at room temperature. Cell extracts were then pelleted. Protein contents of cell lysates were determined using the DC protein assay (Biorad) with BSA as calibration standard. To provide cell volume values and cell counts for normalization, parallel dishes were prepared and trypsinized.

For SIL-CE-ESI-MS, cells were seeded into 6-well plates. After 24 h, the medium was removed, cells were washed, then 2 mL of treatment medium was added. Cells were harvested by trypsinization to maximize yield, before extraction with perchloric acid and TiO$_2$ beads. Parallel dishes were prepared to provide cell counts and protein concentration values for normalization.

*Mouse tissues*: Mouse tissues, a kind gift from Prof. Antonella Riccio (MRC Laboratory for Molecular Cell Biology, University College London), were dissected from 3-month-old C57BL/6J females previously sacrificed to collect embryos. Freshly dissected, PBS washed mouse tissues where homogenized in 1 mL of ice cold 1 M perchloric acid using an IKA DI 18 basic blender at max speed for 30 s. Homogenized tissues were centrifuged at $18,000 \times g$ for 10 min and the supernatant subjected to titanium dioxide extraction as described above. All animal experiments were approved by the UCL Animal Welfare and Ethical Review Body and carried out in accordance to appropriate UK Home Office licenses. Mice were housed in a controlled environment with standard 12:12 light–dark cycle, temperature 19–22 °C, and humidity of 40–55%. All procedures relating to animal care and treatment conformed to the Institutional Animal Care and Use Committee at UCL.

*Plants*: Arabidopsis seeds were surface sterilized and sown onto half-strength Murashige and Skoog (MS) medium supplemented with 1% succrose[34]. Plants were grown under 16 h/8 h day/night conditions at 22 °C/20 °C for 14 days. Light was provided by white LEDs ("True daylight", Polyklima). Shoots (150–180 mg, fresh weight) were shock-frozen in liquid nitrogen, homogenized, and immediately resuspended in 1 M perchloric acid (SigmaAldrich). Titanium dioxide purification of inositol phosphates was carried out as described above.

*D. discoideum*: Wild-type amoeba *D. discoideum* AX2, obtained from the Dicty Stock Center (Northwestern University, Chicago, USA), was grown in SIH defined minimal media (Formedium) at 20 °C in a flask with moderate shaking to a cell density of $3–4 \times 10^6$ cells/mL. Twenty million cells were harvested by centrifugation ($1000 \times g$; 5 min), washed in KK2 buffer (20 mM K-Phosphate buffer pH 6.8), resuspended in 500 μL of ice-cold perchloric acid solution (1 M perchloric acid, 5 mM EDTA) and incubated on ice for 10 min, gently mixing the cell suspension every 2 min. The cell suspension was centrifuged ($15,000 \times g$; 5 min at 4 °C) and the supernatant subject to TiO$_2$ purification as described.

*S. cerevisiae*: Wild-type yeast (BY4741) was grown in Complete Supplement Mixture (SCM) media (Formedium) overnight with shaking at 30 °C to logarithmic phase (OD$_{600}$ = 1–3). Forty OD$_{600}$ units were harvested by centrifugation ($1000 \times g$; 5 min), washed with ice-cold water and resuspended in 500 μL of ice-cold perchloric acid solution (1 M perchloric acid, 5 mM EDTA). After adding ~300 μL of acid-washed glass beads (Sigma Aldrich), yeast were vigorously vortexed for 5 min at 4 °C. The lysate was centrifuged ($15,000 \times g$; 5 min at 4 °C) and the supernatant (acid extract) subject to TiO$_2$ purification as described above.

**Inositol phosphate analysis by SAX-HPLC**. Analysis of InsP pathways after [$^3$H]-inositol radiolabeling was carried out as previously described[25]. Briefly, cells were seeded into 6-well plates and grown in the presence of [$^3$H]-inositol for 5 days to ~80% confluence. Treatment with NaF (10 mM) was for 1 h. Cells were then washed with ice-cold PBS and extracted with perchloric acid, and after neutralization processed for SAX-HPLC analysis.

**Western blotting**. Cells were lysed in TX buffer (50 mM HEPES pH 7.4, 1 mM EDTA, 10% glycerol, 1% Triton X-100, 50 mM NaF, 5 mM sodium pyrophosphate) supplemented with protease and phosphatase inhibitor cocktails (Sigma). Lysates were cleared by centrifugation at $18,000 \times g$ for 5 min at 4 °C, and protein concentrations measured by DC Protein Assay (Bio-Rad). Lysates were resolved using NuPAGE 4–12% bis-tris gels (Life Technologies) and proteins transferred to nitrocellulose membranes. Ponceau S solution (0.1% Ponceau S [Sigma] in 1% acetic acid) was used to confirm equal loading. Membranes were blocked for 1 h in 5% non-fat milk in TBS-T (10 mM Tris base, 140 mM NaCl, 0.05% Tween), then blotted for the following primary antibodies at 1:100–1:1000 overnight in 3% milk: ISYNA1 (sc 271830), actin (sc-1616; Santa Cruz). Secondary antibodies acquired from Sigma, anti-goat IgG−peroxidase conjugated (A5420) and anti-mouse IgG−peroxidase conjugated (A9044) were diluted in 3% milk. The signal was detected using Luminata Crescendo Western Substrate (Merck Millipore) and Amersham Hyperfilm (VWR) and a film developer.

**Reporting Summary**. Further information on research design is available in the Nature Research Reporting Summary linked to this article.

## Data availability
The raw mass spectrometric data files and all other relevant data supporting the findings of this study are available from the corresponding authors upon reasonable request. Source data are provided with this paper.

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

## Acknowledgements

The authors thank Prof. Antonella Riccio for donating mouse tissues. This study was supported by the Deutsche Forschungsgemeinschaft (DFG) under Germany's Excellence Strategy (CIBSS, EXC-2189, Project ID 390939984). This project has received funding from the European Research Council (ERC) under the European Union's Horizon 2020 research and innovation program (grant agreement no. 864246, to H.J.J.). Moreover, this study was supported by a grant from the DFG (DFG Grant JE 572/4-1). We gratefully acknowledge financial support from the Studienstiftung des deutschen Volkes. D.Q. gratefully acknowledges financial support from the Brigitte-Schlieben-Lange-Programm. M.S.W. and A.S. appreciate Medical Research Council (MRC) award MR/T028904/1. D.F. and R.K.H. gratefully acknowledge funding from the Leibniz-Gemeinschaft (SAW-2017-FMP-1). C.G. and S.B.S are funded by the Intramural Program of the National Institute of Environmental Sciences/NIH.

## Author contributions

D.Q., M.S.W., A.S., and H.J.J. conceived the research project and prepared the manuscript with input from all authors. D.Q. developed the CE-MS method and analyzed all samples. M.S.W. and A.S. provided *D. discoideum*, *S. cerevisiae*, and HCT116$^{UCL}$-related samples. V.B.E. prepared HCT116$^{NIH}$-related samples. R.K.H and D.F. supported isotopic internal standards. E.R. and G.S. provided the plant samples. T.M.H., C.W., and N.J. synthesized InsP standards and adjusted their concentrations by NMR. C.G. and S.B.S. provided the HCT116$^{NIH}$ *PPIP5K*$^{-/-}$ cell line. B.K. supported data analysis.

## Funding

## Competing interests

The authors declare no competing interests.
