## [Peer Review File · Nature Communications]

REVIEWER COMMENTS

Reviewer #1 (Remarks to the Author):

The major claim of the submission was the development of a CE-ESI-MS method to characterize the different isomers of myo-inositol phosphates (InsPs) and myo-inositol pyrophosphates (PP-InsPs). Using SIL ISs, the authors demonstrated clearly that CE was effective to separate the isomers, which were then characterized by MS.

I thoroughly enjoyed reading this paper. There are some aspects outside of my specialization, but the chromatography section was carried out extremely well and expertly presented.

Some minor points ...

Throughout ... please ensure that after an abbreviation is defined, it's not redefined again in the paper. This has happened in a few places ... specifically, look for SIL, IS, and BGE usages. As the Methods section is at the end of the paper, I assume that redefining the abbreviations is not necessary.

I would use the term quantitation throughout, and not quantification. In colloquial English the terms "quantitate" and "quantify" are used interchangeably. The main difference is that "quantify" is a somewhat broader term, including classification into broad categories such as "low", "medium" or "high". However, in scientific writing they can carry quite different meanings. In most chemistry work, measurement of quantities involves accuracy, so "quantitate" is the preferred term. Furthermore, "quantification" can also describe the process of developing a mathematical treatment for measuring units. Also, because chemists always use the term "quantitative analysis", for the sake of consistency it is better to use "quantitate" and its derivations.

Pg 1, ln 39 ... I would use hydroxyl here instead of hydroxy, with the latter being typically employed for -OH moieties attached to an aromatic ring.

Pg 2, ln 8 ... a reference is needed for this statement.

Pg 2, ln 25 ... I would start a new paragraph for "Recently ...

Pg 5, ln 15 ... WT cells? Only place I find this WT abbreviation.

Pg 11, ln 1 ... klsamples?

Impressive/convincing study about characterizing the isomers of InsPs, which is novel and will be of interest to others ... I would accept this paper, subject to minor edits.

Reviewer #2 (Remarks to the Author):

My review is not on the biological aspects of the inositol phosphate metabolism, but rather assessing the technical and analytical quality of the CE-ESI-MS method and data.

Novelty: The described CE-ESI-MS is a significant improvement compared to the previously described CE methods, and the LC-MS methods, including the one used as a comparison in the submitted work. More over, the negative ion mode used by the authors allowed the direct analysis of these compounds without the need to labelling, which in the case of the inositol phosphate, could lead to significant structural alterations and other complications. The time required for analysis, peak shape, robustness of the methods all have advantages compared to LC-MS methods, showcasing the suitability of CE-MS for strongly polar compounds and for structural isomers.

Sensitivity: The sensitivity, as described, is acceptable at least for the biological questions addressed. The Agilent CE-MS interface is known for its robustness so for this work it is adequate. In the future when more sensitive methods become available, with 30 to 100 time more sensitive methods, much more information can be obtained. It can be expected that much insights will be obtained by the CE-MS technology.

Robustness of the technology described: I carefully inspected the data presented, in terms of peak shapes and error bars. Some of the peaks are not as symmetrical as one normally expect, but they are very characteristic of peaks produced in CE when the analytes are highly charged. The error bars are very reasonable and are typical of good quality CE-MS work.

Methods described: The methods including sample preparation and CE-ESI-MS, as well as HPLC-MS are well described, and should be easily reproduced by others.

Weakness: The concentration limit of detection was missing in the description also it can be estimated from the given LOQ value. Was the LOQ value 10 times STD? The LOQ were said to be 250 nM. Was it for the best inositol phosphate? How many types of inositol phosphates were monitored? Would it be possible to make a table to show the number of inositol phosphates monitored and their concentrations measured?

David D. Y. Chen

Professor

Department of Chemistry

University of British Columbia

Reviewer #3 (Remarks to the Author):

This is an excellent paper which describes the development of a novel transformative technology with which to measure highly phosphorylated inositol species.

the technology is new, is interesting to the signalling field and certainly deserves publication . the experiments, as expected with this author list, have been carried out to an exceptionally high standard.

Minor issues.

1. the manuscript would benefit from a cartoon of the work flow required for the procedure.
2. it was not clear if the technology would be useful to measure IP3 and IP4 in cell samples and thus would be of direct interest to the lets state more canonical inositol signalling fraternity. could the authors demonstrate an experiment to stimulate PLC activation in cells and then measure IP3 and ip4 signalling.
3. the data presented in figure 5 and 6 suggest that a pool of inositol is synthesised endogenously from glucose and contributes significantly to pools of phosphorylated inositol at low concentrations of inositol often used by experimenters using radiolabelled inositol. this pool is not completely depleted in the ISYNA1 knockout cells, suggesting an alternative pathway for its synthesis. could the authors carry out this C13 inositol labelling in the absence of any additional glucose but with an

alternative substrate to maintain cells. this should show that the higher inositol pool can be labelled to equilibrium faster. this would be relevant to researchers studying this type of signalling.

Prof. Dr. Henning J. Jessen

Director
Institute of Organic Chemistry
Bioorganic Chemistry
Albert-Ludwigs-University
Albertstrasse 21, D-79104 Freiburg

Tel. 0761/203-6073

Tel. 0761/203-6041
Secretary: Regine Schandera

RE: Revision of NCOMMS-20-32877-T

6th October 20

Dear Reviewers,

henning.jessen@oc.uni-freiburg.de
<http://www.jessen-lab.uni-freiburg.de>

on behalf of all authors, I would like thank you for your positive assessment of our work and the time you have invested to evaluate our manuscript. Below you will find a point-to-point response to the comments.

Reviewer #1 (Remarks to the Author):

The major claim of the submission was the development of a CE-ESI-MS method to characterize the different isomers of myo-inositol phosphates (InsPs) and myo-inositol pyrophosphates (PP-InsPs). Using SIL ISs, the authors demonstrated clearly that CE was effective to separate the isomers, which were then characterized by MS.

I thoroughly enjoyed reading this paper. There are some aspects outside of my specialization, but the chromatography section was carried out extremely well and expertly presented.

Some minor points ...

Throughout ... please ensure that after an abbreviation is defined, it's not redefined again in the paper. This has happened in a few places ... specifically, look for SIL, IS, and BGE usages. As the Methods section is at the end of the paper, I assume that redefining the abbreviations is not necessary.

- **We thank the reviewer for this comment. The abbreviations have been harmonized throughout.**

I would use the term quantitation throughout, and not quantification. In colloquial English the terms "quantitate" and "quantify" are used interchangeably. The main difference is that "quantify" is a somewhat broader term, including classification into broad categories such as "low", "medium" or "high". However, in scientific writing they can carry quite different meanings. In most chemistry work, measurement of quantities involves accuracy, so "quantitate" is the preferred term. Furthermore, "quantification" can also describe the process of developing a mathematical treatment for measuring units. Also, because chemists always use the term "quantitative analysis", for the sake of consistency it is better to use "quantitate" and its derivations.

- **We now use the term quantitate/quantitation throughout**

Pg 1, ln 39 ... I would use hydroxyl here instead of hydroxy, with the latter being typically employed for -OH moieties attached to an aromatic ring.

- **We have replaced hydroxyl with OH; to avoid this discussion, as some refer to hydroxyl as the OH radical.**

Pg 2, ln 8 ... a reference is needed for this statement.

- **Two references (2,4) have been inserted.**

Pg 2, ln 25 ... I would start a new paragraph for "Recently ...

- **We have started a new paragraph.**

Pg 5, ln 15 ... WT cells? Only place I find this WT abbreviation.

- **We have replaced WT with wild type.**

Pg 11, ln 1 ... klsamples?

- **The spelling mistake has been corrected.**

Impressive/convincing study about characterizing the isomers of InsPs, which is novel and will be of interest to others ... I would accept this paper, subject to minor edits.

Reviewer #2 (Remarks to the Author):

My review is not on the biological aspects of the inositol phosphate metabolism, but rather assessing the technical and analytical quality of the CE-ESI-MS method and data.

Novelty: The described CE-ESI-MS is a significant improvement compared to the previously described CE methods, and the LC-MS methods, including the one used as a comparison in the submitted work. More over, the negative ion mode used by the authors allowed the direct analysis of these compounds without the need to labelling, which in the case of the inositol phosphate, could lead to significant structural alterations and other complications. The time required for analysis, peak shape, robustness of the methods all have advantages compared to LC-MS methods, showcasing the suitability of CE-MS for strongly polar compounds and for structural isomers.

Sensitivity: The sensitivity, as described, is acceptable at least for the biological questions

addressed. The Agilent CE-MS interface is known for its robustness so for this work it is adequate. In the future when more sensitive methods become available, with 30 to 100 time more sensitive methods, much more information can be obtained. It can be expected that much insights will be obtained by the CE-MS technology.

- **We thank the reviewer and agree that the future for this method is bright.**

Robustness of the technology described: I carefully inspected the data presented, in terms of peak shapes and error bars. Some of the peaks are not as symmetrical as one normally expect, but they are very characteristic of peaks produced in CE when the analytes are highly charged. The error bars are very reasonable and are typical of good quality CE-MS work.

Methods described: The methods including sample preparation and CE-ESI-MS, as well as HPLC-MS are well described, and should be easily reproduced by others.

Weakness: The concentration limit of detection was missing in the description also it can be estimated from the given LOQ value. Was the LOQ value 10 times STD? The LOQ were said to be 250 nM. Was it for the best inositol phosphate? How many types of inositol phosphates were monitored? Would it be possible to make a table to show the number of inositol phosphates monitored and their concentrations measured?

- **The LOD and LOQ values are obtained based on Signal-to-Noise Approach. We have clarified LOD/LOQ in the manuscript and also added a table with the requested values (Supplementary figure 2a) and an explanation/example (Supplementary figure 2b) in the SI. In the paper, the statement has been changed to “The limits of quantitation (LOQs) for different InsPs were 150-500 nM (Supplementary Figure 2)”**

Reviewer #3 (Remarks to the Author):

This is an excellent paper which describes the development of a novel transformative technology with which to measure highly phosphorylated inositol species.

The technology is new, is interesting to the signalling field and certainly deserves publication. the experiments, as expected with this author list, have been carried out to an exceptionally high standard.

- **We thank the reviewer for these very positive comments.**

Minor issues.

1. the manuscript would benefit from a cartoon of the work flow required for the procedure.

- **We agree with the reviewer and have added a workflow graphic to the SI (Supplementary Figure 4)**

2. it was not clear if the technology would be useful to measure IP3 and IP4 in cell samples and thus would be of direct interest to the lets state more canonical inositol signalling fraternity. could the authors demonstrate an experiment to stimulate PLC activation in cells and then measure IP3 and ip4 signalling.

- **We agree with the reviewer that IP3 and IP4 isomers are also of interest. To demonstrate that such an analysis is in principle possible, the original submission (fig. 2b) already contained electropherograms for three different IP4s, clearly demonstrating the potential of CE-MS to also monitor these isomers. The current conditions (optimized with a BGE for higher inositol phosphates from IP5 to IP8) do not enable baseline separation of IP3 and IP4, so IP3 measurements are difficult due to potential neutral loss of phosphate from IP4 (producing IP3). While this does not affect IP4 analysis, for the analysis of IP3 a different method will have to be developed, which is a significant future endeavor.**
- **We do not (yet) have stable isotopic standards for both IP3 (20 possible isomers) and IP4 (15 possible isomers), which will be a goal for future studies but significantly beyond the scope of this study. We are confident that CE-MS will be very useful also for the analysis of these isomers. However, the current work focuses on quantitations of the higher inositol phosphates with stable isotope standards, which will take years to produce for IP3 and IP4 for high quality analyses. Therefore, the experiments suggested are not currently feasible to the same high standard of the presented studies herein.**

3. the data presented in figure 5 and 6 suggest that a pool of inositol is synthesised endogenously from glucose and contributes significantly to pools of phosphorylated inositol at low concentrations of inositol often used by experimenters using radiolabelled inositol. this pool is not completely depleted in the ISYNA1 knockout cells, suggesting an alternative pathway for its synthesis. could the authors carry out this C13 inositol labelling in the absence of any additional glucose but with an alternative substrate to maintain cells. this should show that the higher inositol pool can be labelled to equilibrium faster. this would be relevant to researchers studying this type of signalling.

- **We are pleased that the reviewer recognizes the biological implications of an alternative pathway for inositol synthesis. The reviewer suggests labelling cells with ¹³C-inositol after substituting glucose with an alternative carbon source to detect a faster synthesis of higher phosphorylated ¹³C-inositol phosphates. Yet, glucose represents the starting point of general metabolism in animal cells. Substituting glucose with fructose (or with galactose for a shorter time) while possible would induce unwanted changes in cell growth and metabolism (PMID: 6500609), thereby precluding the meaningful characterization of the kinetic of ¹³C-inositol incorporation in inositol phosphates. Furthermore, HCT116 cells can perform gluconeogenesis**

(PMID: 25009184 and 28655758) to generate glucose from intermediate metabolites, preventing us from properly performing the suggested experiment. Nevertheless, we would like to reassure the reviewer that our efforts are concentrated on identifying the alternative pathway for inositol synthesis that our newly developed CE-MS technology has unexpectedly revealed and to learn more about the kinetics of endogenous synthesis.

I would like to thank the reviewers again for their valuable input and also for sharing our excitement about this method. Indeed, it will now be possible to address many open questions in the InsP signalling field.

With kind regards,

Henning Jessen